# Traumatic brain injury and risk of heart failure and coronary heart disease: A nationwide population-based cohort study

**Ching-Hui Huang[1], Chao-Tung Yang[2,3]☯ \*, Chia-Chu Chang ●[4,5]☯ \***

**1** Division of Cardiology, Department of Internal Medicine, Changhua Christian Hospital, Changhua, Taiwan,
**2** Department of Computer Science, Tunghai University, Xitun District, Taichung City, Taiwan, **3** Research Center for Smart Sustainable Circular Economy, Tunghai University, Xitun District, Taichung City, Taiwan, **4** Department of Internal Medicine, Kuang Tien General Hospital, Taichung, Taiwan, **5** Department of Nutrition, Hungkuang University, Taichung, Taiwan

☯ These authors contributed equally to this work.
\* chiachuchang0312@gmail.com (CCC); ctyang@thu.edu.tw (CTY)

## Abstract

### Background

This study examined the long-term risks of heart failure (HF) and coronary heart disease (CHD) following traumatic brain injury (TBI), focusing on gender differences.

### Methods

Data from Taiwan's National Health Insurance Research Database included 29,570 TBI patients and 118,280 matched controls based on propensity scores.

### Results

The TBI cohort had higher incidences of CHD and HF (9.76 vs. 9.07 per 1000 person-years; 4.40 vs. 3.88 per 1000 person-years). Adjusted analyses showed a significantly higher risk of HF in the TBI group (adjusted hazard ratio = 1.08, 95% CI = 1.01–1.17, P = 0.031). The increased CHD risk in the TBI cohort became insignificant after adjustment. Subgroup analysis by gender revealed higher HF risk in men (aHR = 1.14, 95% CI = 1.03–1.25, P = 0.010) and higher CHD risk in women under 50 (aHR = 1.32, 95% CI = 1.15–1.52, P < 0.001). TBI patients without beta-blocker therapy may be at increased risk of HF.

### Conclusion

Our results suggest that TBI increases the risk of HF and CHD in this nationwide cohort of Taiwanese citizens. Gender influences the risks differently, with men at higher HF risk and younger women at higher CHD risk. Beta-blockers have a neutral effect on HF and CHD risk.

**Data Availability Statement:** Raw Data cannot be shared publicly due to the government regulation to protect data privacy and security. Data are available from the Taiwan Health and Welfare Data Centre (HWDC) for researchers who meet the

criteria for access to confidential data. HWDC, a data repository site that centralizes the NHIRD databases for data management and analyses, and use of the data must be for research purposes only. All applications are subject to expert review to ensure reasonable use. Applicants must obtain IRB approval before applying to access the NHIRD. Currently, only Taiwanese researchers have direct access to the data. The authors remotely accessed the data from the data centre of the Ministry of Health and Welfare in Taiwan. Researchers interested in accessing this dataset could submit a formal application to the Taiwan Ministry of Health and Welfare to request access (No 488, Sect. 6, Zhongxiao E Rd, Nangang District, Taipei 115, Taiwan; website: https://dep.mohw.gov.tw/DOS/cp-2516-59203-113.html). All relevant data are within the manuscript and the minimal dataset is contained within our paper.

**Funding:** The author(s) received no specific funding for this work.

**Competing interests:** The authors have declared that no competing interests exist.

## Introduction

Traumatic brain injury (TBI) is a serious public health problem worldwide associated with an increased risk of disability and mortality [1]. Brain-heart interactions are most common in traumatic brain injury and manifest as arrhythmias, neurogenic myocardial stunning or stress cardiomyopathy, hemodynamic disturbances, and death [2, 3]. In most patients with stress cardiomyopathy, cardiac function may return within hours to 6–12 weeks, with or without cardiac intervention. However, patients with stress cardiomyopathy remain at risk of recurrence even years after the initial event [4] and there are limited data on the incidence of these interactions (i.e., heart failure after TBI). Previous studies have focused on associations with the risk of coronary heart disease (CHD) after TBI, while little information is available on the incidence of heart failure (HF) after TBI [5, 6]. Few studies have simultaneously investigated CHD and HF after TBI. Furthermore, women are prone to stress cardiomyopathy [7]. However, the long-term follow-up risks of HF and CHD by gender in patients with TBI are unknown.

Brain-heart crosstalk, which occurs due to stress-induced surges in catecholamines after brain injury and brain injury-induced release of inflammatory mediators, can further damage cardiomyocytes and affect patient clinical outcomes [8, 9]. Since sympathetic system overactivity is believed to play a key role in the cardiac manifestations of TBI, several studies have investigated the protective role of beta-blockers in this setting [3]. A recent meta-analysis demonstrated that beta-blocker use after TBI reduces in-hospital mortality [10]. However, it is unclear whether the use of beta-blockers in patients with TBI reduces the risk of subsequent HF.

A better understanding of the interaction between TBI and the development of cardiovascular comorbidities may have important implications for preventive care, prognosis, and targeted screening of high-risk populations. The purpose of our study was to evaluate whether the incidence of CHD and HF is higher in a cohort of TBI than in controls and whether there are gender differences. Furthermore, it was explored whether the use of beta-blockers in patients with TBI may reduce the risk of subsequent HF.

## Materials and methods

### Data source

Patients' data were retrieved from Taiwan's National Health Insurance Research Database (NHIRD), which contains health-care claims data from the National Health Insurance (NHI) program from 1996 to 2012. The claims in NHIRD include the demographics, clinical visit records, ambulatory care, hospital admissions, disease status, drug prescriptions, medical procedures and dental services of approximately 23 million people (>99% of Taiwan's population) [11]. Diseases were identified by the International Classification of Diseases, Ninth Revision, Clinical Modification (ICD-9-CM) codes. Because the analyzed information was de-identified and encrypted, the need for consent was waived by the ethics committee and was approved by the Institutional Review Board of the Changhua Christian Hospital (approval number 190127).

### Study population

We identified newly diagnosed TBI patients (ICD-9-CM codes 800, 801, 803, 804, 850–853, 854.1, and 959.01) from January 1 1996 to December 31 2012. The index date was defined as the first date of TBI diagnosis. Patients who had developed heart failure (ICD-9-CM codes 428.0, 429.83) and coronary heart disease (ICD-9-CM codes 410–414) before the index date, those aged <18 years, those who survived for < 30 days after TBI diagnosis, those who were followed for < 30 days, and those with incomplete demographic information were excluded

from the study. Each identified TBI patient was matched with four control subjects by propensity scoring. The control cohort comprised patients without a history of CHD, heart failure, or TBI before the index date. A flowchart of the cohort selection process is presented in Fig 1. To balance the measured covariates between the two study cohorts, we calculated the propensity score by using multivariate logistic regression to predict the likelihood of TBI for each patient. For each patient with TBI, one control without TBI was selected through matching by age, calendar year of index date, and propensity score. We used the nearest-neighbor algorithm with a caliper of 0.1 SD units to generate matched pairs. Details of the propensity score model are described in our previous work [12].

## Outcome measures and relevant variables

We examined ICD-9-CM codes in the patients' records to determine outcomes and comorbidities. Both the TBI and control cohorts were followed up from the date of study enrollment to the date of the first occurrence of CHD or HF, date of withdrawal from the NHI program, or end of 2013. The date of withdrawal from the NHI program has been recognized as an accurate and reliable proxy for the date of death in Taiwan [13]. CHD was identified on the basis of the corresponding diagnostic codes (ICD-9-CM 410–414) recorded for at least three outpatient clinic visits or at least one hospital admission. HF were identified on the basis of the diagnostic code (ICD-9-CM 428.0, 429.83) recorded for at least three outpatient clinic visits or at least one hospital admission. All data on comorbidities, including hypertension, hyperlipidemia, diabetes mellitus, chronic kidney disease, stroke, chronic obstructive pulmonary disease, peripheral artery occlusive disease, and dysrhythmia, were collected from the NHIRD. Major comorbid diseases diagnosed in at least three records in the claims data 1 year before the index date were defined as baseline comorbidities. Charlson comorbidity index (CCI) scores were used to quantify baseline comorbidities. The CCI is the most extensively applied comorbidity index for predicting mortality [14]. It identifies the comorbidities of patients and weights them based on the adjusted risk of mortality or health resource use; the sum of all the weights then provides a single comorbidity score. A score of 0 indicates that no comorbidities were present in patients. A higher score indicates that the predicted outcome will more likely result in a higher risk of mortality or higher health resource use. Additionally, long-term medication use is believed to be associated with CHD or HF outcomes in TBI patients, and in this study, the long-term use of medications, namely antidiabetic agents, statins, antihypertension medications (e.g., angiotensin-converting enzyme inhibitors or angiotensin II receptor blockers), beta-blockers, and diuretics were recorded according to the Anatomical Therapeutic Chemical codes defined by the World Health Organization. Patients who used these in the follow-up period, the corresponding data were censored. Patients who used these medications for more than 90 consecutive days prior to receiving a diagnosis of CHD or HF were defined as medication users. If the medications were stopped for more than 180 days during the follow-up period, the corresponding data were censored.

## Statistical analysis

The demographic data and other clinically relevant data of both cohorts are presented as proportions and means ± SD. A standardized difference (StD) of more than 0.1 indicated significant heterogeneity between the two cohorts. The Cox proportional hazard model was used to estimate the relative risk of CHD or HF in the TBI and control cohorts. Adjusted hazard ratios (aHRs) were estimated using multivariate Cox analysis, which was adjusted for confounders, namely age, CCI score, monthly income, comorbidities, and long-term medication use. Because comorbidities may not only be present at baseline but also develop during follow-up,

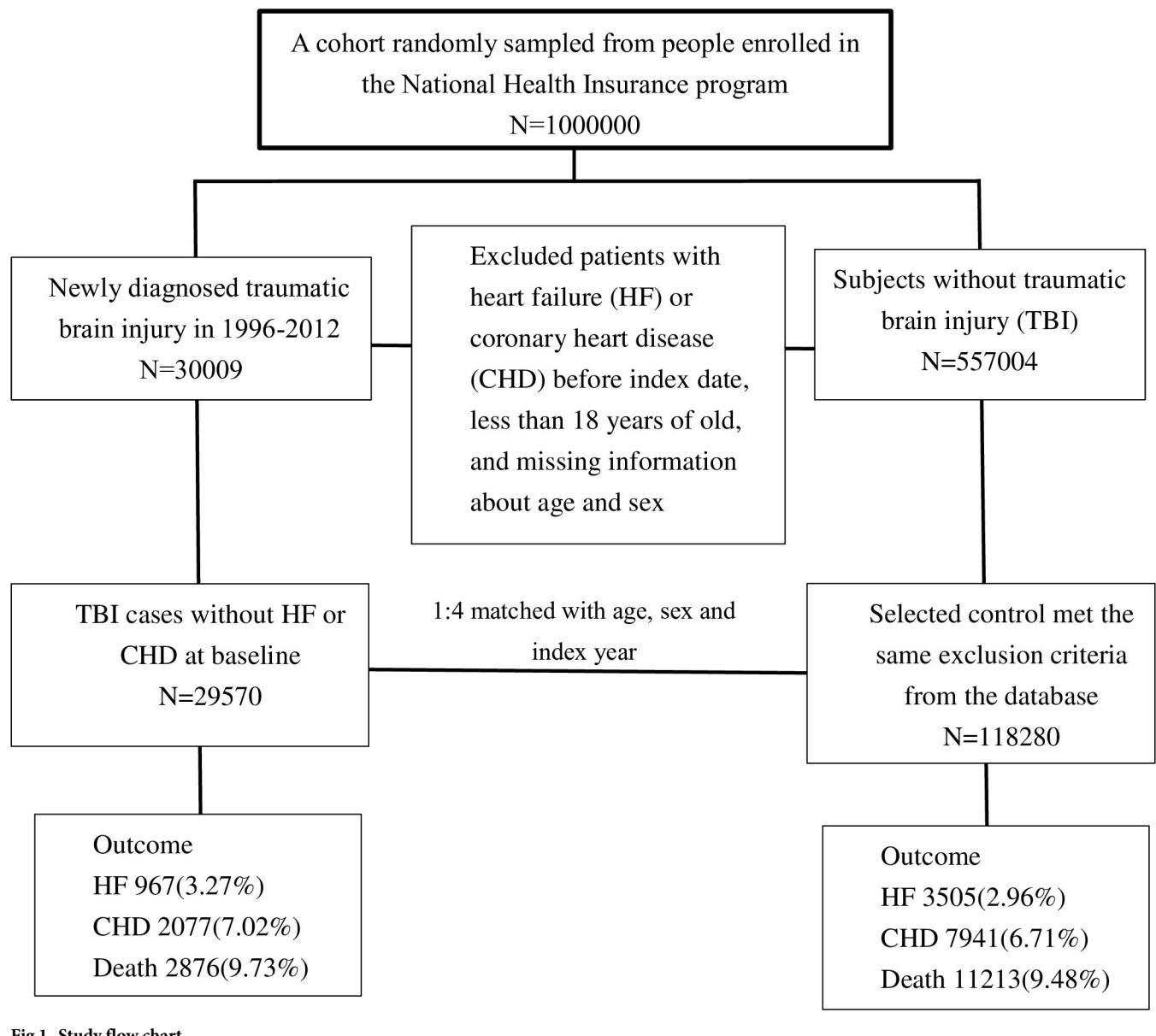

**Fig 1. Study flow chart.**

comorbidities were modeled using nonreversible time-dependent binary covariates for event analyses. A competing risk is an event that either hinders the development of the event of interest or modifies the chance of the occurrence of this event. Because CHD and HF risks might compete with the risk of death, the Cox proportional hazard model with competing risks (Fine–Gray model) of death was used to estimate the relative risk of CHD or HF in the TBI cohort compared with the control cohort. This model is a proportional hazards model employed for subdistribution analysis. In this study, a test of interaction was conducted to determine observable subgroup effects. All statistical analyses were performed using SAS 9.4 software (SAS Institute Inc., Cary, NC, USA). Two-tailed P values of <0.05 were considered statistically significant.

## Results

### Characteristics of patients

Table 1 lists the baseline demographic data of enrolled patients. This study enrolled 587,010 participants, including 30,009 patients diagnosed with TBI initially, and finally identified 29,570 patients after propensity score-matched diagnosed with TBI without HF or CHD at baseline, and 118,280 propensity score-matched controls not diagnosed with TBI and without HF or CHD at baseline (Fig 1). The mean follow-up times of these cohorts were 7.75±4.52 and 7.55±3.92 years, respectively. The average age, CCI score, sex distribution, and proportions of patients with hypertension, hyperlipidemia, CAD, cardiac arrhythmia, stroke, PAOD, and drug usage were similar in both cohorts after propensity score-matched (Table 1).

### Incidence and risk of HF in TBI and control cohorts

During the follow-up period, the number of HF events in the control and TBI cohorts was 3505 (3.88 per 1000 person-years) and 967 (4.40 per 1000 person-years), respectively (Table 2). Kaplan–Meier analysis revealed that the cumulative incidence of HF was significantly different in the cohorts (log-rank test, P = 0.001) (Fig 2A). Three models were employed to adjust the risk of HF in the TBI cohort in comparison with that in the control cohort (Table 2). In Model 1, after propensity score matching, the risk of HF was significantly higher in the patients with TBI than in the controls (crude HR = 1.13, 95% confidence interval [CI] = 1.05–1.21, P = 0.001). In Model 2, after propensity score adjustment, the incidence of HF remained higher in the TBI cohort than in the control cohort (aHR = 1.11, 95% CI = 1.04–1.20, P = 0.004). In Model 3, after adjustment for all the confounders listed in Table 1, the risk of HF remained higher in the TBI cohort than in the control cohort (aHR = 1.08, 95% CI = 1.01–1.17, P = 0.031).

### Incidence and risk of CHD in TBI and control cohorts

During the follow-up period, the number of CHD events in the control and TBI cohorts was 7941 (9.07 per 1000 person-years) and 2077 (9.76 per 1000 person-years), respectively (Table 2). Kaplan–Meier analysis revealed that the cumulative incidence of CHD was significantly different in the cohorts (log-rank test, P = 0.003) (Fig 2B). Three models were employed to adjust the risk of CHD in the TBI cohort in comparison with that in the control cohort (Table 2). In Model 1, after propensity score matching, the risk of CHD was significantly higher in the patients with TBI than in the controls (crude HR = 1.07, 95% confidence interval [CI] = 1.02–1.13, P = 0.005). In Model 2, after propensity score adjustment, the incidence of CHD remained higher in the TBI cohort than in the control cohort (aHR = 1.06, 95% CI = 1.01–1.11, P = 0.021). In Model 3, after adjustment for all the confounders listed in Table 1, the risk of CHD was not significantly different in the TBI cohort than in the control cohort (aHR = 1.03, 95% CI = 0.98–1.08, P = 0.237).

### Subgroup analyses of the risk of CHD or HF between TBI and control cohorts

Table 3 lists the subgroup analyses of the risk of CHD and HF between the TBI and control cohorts. Subgroup analyses were performed to determine the association between TBI and the risk of CHD or HF for the age groups of <40 years and > = 40 years, subgroups without and with comorbidity, subgroups without and with beta-blocker used in both cohorts. The results revealed that age > = 40 years will increase the risk of HF (aHR = 1.19. 95% CI = 1.10–1.28, P< 0.001). Both age groups (<40 years and > = 40 years) will increase the risk of CHD. The

**Table 1. Patient's characteristics.**

| Characteristics | Overall patients | | | Propensity score matching | | |
|---|---|---|---|---|---|---|
| | non-TBI cohort (n = 557004) | TBI cohort (n = 30009) | StD | non-TBI cohort (n = 118280) | TBI cohort (n = 29570) | StD |
| *Demographics* | | | | | | |
| Age, years | 40.52±14.89 | 42.56±18.25 | 0.123 | 41.46±16.78 | 42.32±18.16 | 0.049 |
| Male gender | 267193(47.97%) | 16799(55.98%) | 0.161 | 66535(56.25%) | 16470(55.7%) | 0.011 |
| Monthly income, NTD | 19262.34±16368.88 | 14997.82±12007.86 | 0.297 | 15407.08±14110.13 | 15103.55±12033.17 | 0.023 |
| Geographic location | | | | | | |
| Northern Taiwan | 285221(51.21%) | 11860(39.52%) | 0.236 | 46923(39.67%) | 11782(39.84%) | 0.004 |
| Central Taiwan | 101573(18.24%) | 5910(19.69%) | 0.037 | 23182(19.6%) | 5824(19.7%) | 0.002 |
| Southern Taiwan | 158122(28.39%) | 11409(38.02%) | 0.206 | 45125(38.15%) | 11152(37.71%) | 0.009 |
| Eastern Taiwan and islands | 12088(2.17%) | 830(2.77%) | 0.038 | 3050(2.58%) | 812(2.75%) | 0.010 |
| Clinic visit frequency (visits per year) | 11±11.44 | 14.96±14.66 | 0.302 | 14.17±14.11 | 14.47±13.62 | 0.021 |
| *Comorbidities* | | | | | | |
| CCI score | 0.54±1.04 | 0.85±1.38 | 0.249 | 0.77±1.28 | 0.82±1.35 | 0.040 |
| Hypertension | 45984(8.26%) | 3994(13.31%) | 0.163 | 14143(11.96%) | 3758(12.71%) | 0.023 |
| Hyperlipidemia | 24505(4.4%) | 1747(5.82%) | 0.065 | 6414(5.42%) | 1681(5.68%) | 0.011 |
| DM | 21753(3.91%) | 2170(7.23%) | 0.145 | 7716(6.52%) | 2031(6.87%) | 0.014 |
| Gout | 13195(2.37%) | 1081(3.6%) | 0.073 | 3878(3.28%) | 1024(3.46%) | 0.010 |
| COPD | 7238(1.3%) | 839(2.8%) | 0.106 | 2792(2.36%) | 754(2.55%) | 0.012 |
| Stroke | 3499(0.63%) | 596(1.99%) | 0.120 | 1797(1.52%) | 509(1.72%) | 0.016 |
| AF | 515(0.09%) | 68(0.23%) | 0.034 | 212(0.18%) | 62(0.21%) | 0.007 |
| PAOD | 984(0.18%) | 104(0.35%) | 0.033 | 351(0.3%) | 93(0.31%) | 0.003 |
| *Long-term medications* | | | | | | |
| dm drug | 14489(2.6%) | 1360(4.53%) | 0.104 | 4924(4.16%) | 1291(4.37%) | 0.010 |
| Anti-hypertensive drugs | 34377(6.17%) | 2650(8.83%) | 0.101 | 9675(8.18%) | 2532(8.56%) | 0.014 |
| ACEIs/ARBs | 16181(2.91%) | 1262(4.21%) | 0.070 | 4540(3.84%) | 1215(4.11%) | 0.014 |
| Diuretics | 8163(1.47%) | 711(2.37%) | 0.066 | 2584(2.18%) | 677(2.29%) | 0.007 |
| Beta-blockers | 19910(3.57%) | 1339(4.46%) | 0.045 | 4943(4.18%) | 1297(4.39%) | 0.010 |
| Statins | 11428(2.05%) | 839(2.8%) | 0.048 | 3126(2.64%) | 820(2.77%) | 0.008 |
| Propensity score | 0.05±0.03 | 0.08±0.06 | 0.553 | 0.07±0.05 | 0.07±0.05 | 0.000 |
| *Outcome* | | | | | | |
| HF | 9831(1.76%) | 1050(3.5%) | 0.108 | 3505(2.96%) | 967(3.27%) | 0.018 |
| CHD | 24808(4.45%) | 2230(7.43%) | 0.126 | 7941(6.71%) | 2077(7.02%) | 0.012 |
| Death | 46133(8.28%) | 3025(10.08%) | 0.062 | 11213(9.48%) | 2876(9.73%) | 0.008 |
| Follow-up time (year) | 6.77±3.89 | 7.59±3.93 | 0.211 | 7.75±4.52 | 7.55±3.92 | 0.046 |

ACEIs: Angiotensin Converting Enzyme inhibitors; ARBs: Angiotensin II Receptor Blockers; AF: Atrial Fibrillation; CCI score: Charlson Comorbidity Index score; CHD: Coronary Heart Disease; COPD: Chronic Obstructive Pulmonary Disease; DM: Diabetes Mellitus; HF: Heart Failure; NTD: New Taiwan Dollar; PAOD: Peripheral arterial occlusion disease; StD: Standard Deviation; TBI: traumatic brain injury

reference group was the control cohort. Additionally, TBI patients without using beta-blocker will increase the risk of HF (aHR = 1.09. 95% CI = 1.00–1.17, P = 0.004). TBI patients using beta-blocker have neutral effect of HF risk. The risks of incident HF were significantly higher in men (aHR = 1.14, 95% confidence interval [CI] = 1.03–1.25, P = 0.010). The risks of CHD were significantly higher in women under 50 years of age (aHR = 1.32, 95% confidence interval [CI] = 1.15–1.52, P<0.001). In this study, interaction effects were only assessed between TBI and categorical variables.

**Table 2. Incidence and risk of HF or CHD in patients with TBI and matched participants.**

| | Event | incidence‡ | Model 1 | | Model 2 | | Model 3 | |
|---|---|---|---|---|---|---|---|---|
| | | | HR (95% CI) | P-value‡ | Adj.HR (95% CI) | P-value‡ | Adj.HR (95% CI) | P-value‡ |
| *HF* | | | | | | | | |
| Control cohort | 3505 | 3.88 (3.75–4.01) | reference | | reference | | reference | |
| TBI cohort | 967 | 4.40 (4.12–4.68) | 1.13(1.05–1.21) | 0.001 | 1.11(1.04–1.20) | 0.004 | 1.08(1.01–1.17) | 0.031 |
| *CHD* | | | | | | | | |
| Control cohort | 7941 | 9.07 (8.87–9.27) | Reference | | Reference | | Reference | |
| TBI cohort | 2077 | 9.76 (9.34–10.18) | 1.07(1.02–1.13) | 0.005 | 1.06(1.01–1.11) | 0.021 | 1.03(0.98–1.08) | 0.237 |

Model 1: Propensity score matched.

Model 2: Adjusted for propensity score.

Model 3: Adjusted for all variables listed in Table 1.

‡per 1000 person-years. HR: hazard ratio; Adj.HR: adjusted hazard ratio; CI: confidence interval.

‡All analyses incorporated in regard to death as competing risks.

## Discussion

### The incidence of CHF was higher in the TBI cohort than in the control cohort

During the follow-up period, the number of HF events and the cumulative incidence of HF were significantly higher than in the control cohort (Table 2 and Fig 2). The association of TBI with subsequent CHF was not attenuated in multivariate models, suggesting that TBI may account for risk independent of the other variables. Our findings highlight some important points. First, to our best knowledge, our study provided general population data on the incidence of HF following TBI insults. The incidence was lower than that of CHD. Previous large-scale cohort studies focused on composite end points for cardiovascular disease (coronary artery disease, stroke, and peripheral artery disease) in U.S Veterans after 9/11 [5], or with the subsequent incidence of major adverse cardiovascular and cerebrovascular events (MACCE) in the general population. This definition of MACCE include a composite of cardiovascular disease, ischemic stroke, and death [15]. No large-scale study had mentioned the incidence of HF following TBI events. Most patients with stress cardiomyopathy show regression of cardiac function independently of cardiac intervention, ranging from hours to 6–12 weeks [16]. However, patients with stress cardiomyopathy remain at risk of recurrence even years after the initial event [17]. Our data provided information to the knowledge gap to better understand the interaction between TBI and HF development. Second, a long-standing belief is that older women have a higher prevalence of stress cardiomyopathy [4, 16]. However, in our study we found that the risk of incident HF was significantly higher in men than women in the population of TBI, and the mean age of the TBI cohort was 42.56±18.2 years old. Our real-world data contrary to previous belief. Third, previous meta-analyses have shown that beta-blockers are effective in reducing mortality in patients with TBI [10]. Exposure to beta-blockers appears to improve functional outcome in patients with severe TBI in a matched case-control study [18]. However, it is unclear whether the beta-blockers used in patients with TBI reduce the risk of subsequent HF. In our study, we found that treatment of patients with β-blockers offers a neutral effect on the risk of HF and CHD, but patients with TBI who did not receive β-blockers treatment may increase the risk of HF. Our result suggested that the beta-blocker used after TBI might have a beneficial effect on reducing the risk of subsequent incidence of HF. Because

**A: Kaplan-Meier curve for HF**

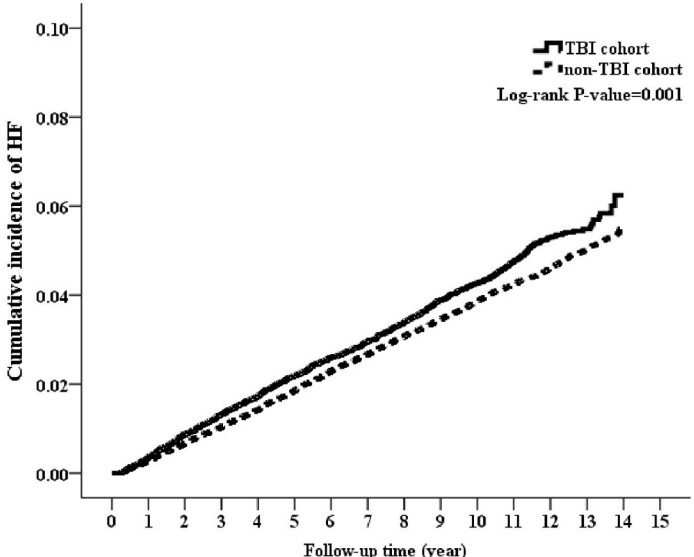

**B: Kaplan-Meier curve for CHD**

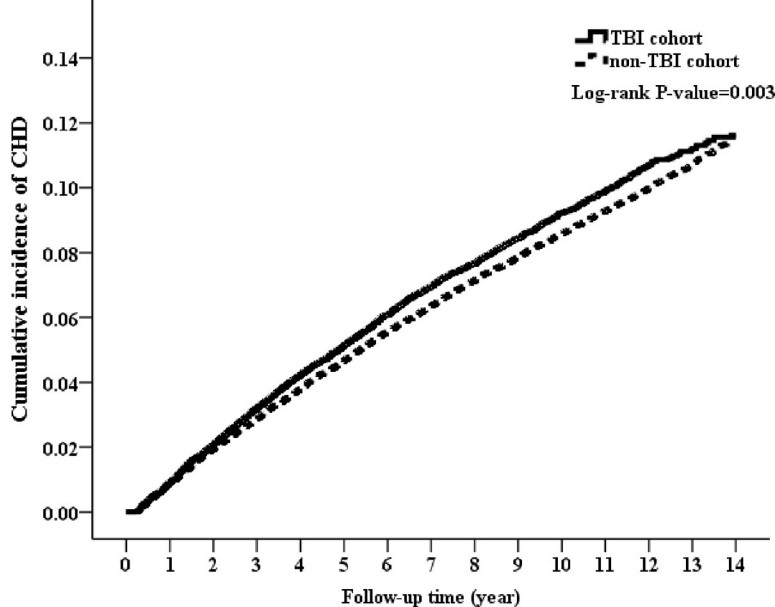

**Fig 2.** A. Kaplan–Meier analysis of cumulative incidence of HF in TBI and control Cohorts. B. Kaplan–Meier analysis of cumulative incidence of CHD in TBI and control Cohorts.

this database is a claim database, we were unable to explore the reasons why patients should or should not use beta-blockers after TBI.

## Age stratification and risk of CHD after TBI

To determine whether the risk of CHD and HF changes with age, we assessed the risk after TBI compared to those younger than or older than 40 years. The 40-year cutoff point was

**Table 3. Subgroup analyses of the risk of HF and CHD in patients with TBI and matched participants.**

| Subgroup | TBI group compared to control group | | | | | |
|---|---|---|---|---|---|---|
| | aHR for HF (95% CI) | P-value | $P_{interaction}$ | aHR for CHD (95% CI) | P-value | $P_{interaction}$ |
| Age, years | | | 0.334 | | | 0.017 |
| <40 | 1.05(0.79–1.39) | 0.764 | | 1.30(1.12–1.51) | 0.001 | |
| ≥540 | 1.19(1.10–1.28) | <0.001 | | 1.07(1.01–1.12) | 0.016 | |
| Comorbidity | | | 0.354 | | | 0.159 |
| 0 | 1.10(0.98–1.23) | 0.103 | | 1.05(0.97–1.12) | 0.226 | |
| ≥1 | 1.06(0.97–1.16) | 0.216 | | 1.02(0.95–1.09) | 0.668 | |
| Beta-blockers | | | 0.755 | | | 0.889 |
| No | 1.09(1.00–1.17) | 0.040 | | 1.03(0.98–1.09) | 0.290 | |
| Yes | 1.07(0.88–1.31) | 0.474 | | 1.08(0.94–1.25) | 0.277 | |
| Gender | | | 0.183 | | | 0.830 |
| Male | 1.14(1.03–1.25) | 0.010 | | 1.02(0.96–1.09) | 0.525 | |
| Female | 1.02(0.91–1.13) | 0.794 | | 1.042(0.97–1.12) | 0.289 | |
| Female aged<50 | 0.96(0.70–1.31) | 0.789 | 0.490 | 1.32(1.15–1.52) | <0.001 | <0.001 |
| Female aged> = 50 | 1.10(0.98–1.24) | 0.099 | | 0.99(0.91–1.08) | 0.853 | |

§ Model was adjusted for all variables listed in Table 1.

All analyses incorporated in regard to death as competing risks.

based on previous large cohort studies [6, 19], both of which showed that younger age groups (18–40 years) had significantly higher risk of cardiovascular disease. Our results indicate an association between TBI and CHD in all age groups without a baseline diagnosis. The risk of CHD was significantly higher in the age group less than 40 years, with an adjusted hazard ratio of 1.30. Our findings are consistent with previous studies showing that TBI of varying severity increases the risk of subsequent CHD, and the risk is significantly higher in patients younger than 40 years of age [6, 19]. Regarding gender, assuming menopause at 50 years old, we found that women with TBI under 50 years old had a significantly higher risk of CHD, which is contrary to what we have long known about the incidence of CHD in menopausal women. Our data are consistent with previous studies showing that TBI is a potential risk factor for CHD even in premenopausal women [5, 6, 19, 20].

Mechanisms of CHD in patients with TBI were complex. An observational study has shown the presence of coronary artery calcification in patients with mild TBI [20]. A preclinical study in mice supports a causal relationship between TBI and atherosclerosis [21], which may be related to increased sympathetic activity. Studies in rodents and humans have found long-lasting increases in sympathetic activity after TBI [22–24]. Beta-adrenergic receptor antagonism reduces accelerated atherosclerosis after TBI, as demonstrated in a mouse model deficient in apolipoprotein E [25]. Sympathoadrenal activation after TBI induces coagulopathy and endotheliopathy, as demonstrated by biomarkers, and these effects are associated with a poor prognosis [26]. A previous study has demonstrated the effects of TBI on leukocyte activation and endothelial adhesion biomarkers [21], although the exact mechanisms of accelerated atherosclerosis remain to be demonstrated. In addition to the systemic effects of catecholamines on circulating monocytes or endothelial cells after brain injury, efferent sympathetic peripheral nervous system axons may be stimulated to produce epinephrine locally in susceptible arteries, leading to increased inflammatory plaque activity [27]. All of the above possible explanations may explain why patients with TBI may be at increased risk for developing CHD.

### TBI severity and subsequent CHD and HF risks

Current tools to assess the severity of TBI include the level of consciousness, typically assessed using the Glasgow Coma Scale (GCS), and the duration of posttraumatic amnesia [28]. The use of GCS scores is more commonly used. TBI can be classified into mild, moderate, and severe injury types based on patient scores reported using the GCS [29]. Mild traumatic brain injuries can result from concussions during sports activities, falls in elderly subjects, and battle-field injuries; while moderate to severe TBI are responsible for the majority of death and disability [29]. The incidence of mild TBI has been reported to be higher than that of moderate to severe TBI [30]. In terms of cardiovascular comorbidities, mild TBI (e.g. concussion) increases the risk of CHD even in younger age groups [19]. Furthermore, in a longitudinal study of individuals with TBI up to 10 years after injury, the authors found that CHD was highly prevalent in subjects with mild TBI, as well as moderate to severe injuries, and that some cardiovascular disease risk was greater in subjects with mild TBI [6]. Data are inconsistent when it comes to the risk of HF and the severity of TBI. A study showed that the severity of head computed tomography images was not associated with the development of cardiac dysfunction [31]. Another observational study showed that traumatic brain injury was not associated with significant myocardial dysfunction [32]. However, a study has reported that previously healthy patients with moderate to severe traumatic brain injury have a higher incidence of early systolic dysfunction than patients with mild traumatic brain injury and that it is reversible within the first week of hospitalization [33]. In our study, we did not emphasize the severity of TBI and the risk of CHD or HF in the absence of consistent arguments. We aimed to use this national database to simultaneously assess long-term, CHD, and HF events in a large and representative sample of patients with TBI compared to a matched cohort.

### Clinical implication

A major focus of TBI management is to limit secondary brain damage. However, some patients with TBI develop extracranial multi-organ dysfunction, leading to secondary brain injury, increased risk of death, and poor functional prognosis after TBI. Although the data source records from 1996 to 2012, that is, 10 years ago. Our study can provide information on the incidence of these interactions (i.e. HF and CHD after TBI) to fill the knowledge gap resulting from the lack of randomized clinical trials and identify susceptible subgroups for screening and intervention to improve the prognosis of the patient. Increase the awareness of the healthcare provider and patient of TBI as a potential risk factor for CHD and HF, highlighting that aggressive control of cardiovascular risk factors is mandatory, especially in young adults after TBI.

### Study limitation

Our study has some limitations. First, the study used a claims database that lacked data on HF-related imaging, such as echocardiography and cardiac magnetic resonance imaging. Therefore, we were unable to determine the HF subtype, nor did we have information on the percentage of HFpEF, the severity of HF (NYHA level), and echocardiographic parameters in patients with HF. Second, NHIRD did not provide some information, such as smoking history, body mass index, physical activity, blood pressure, inflammatory markers, blood sugar, blood lipids, and severity of TBI. Lacking information on the severity of TBI, we were unable to measure whether there were dose-dependent effects on the severity and subsequent risk of CHD or HF. These unmeasured covariates may have influenced the results, even after balancing baseline clinical characteristics and propensity score matching. Third, this study was designed retrospectively; therefore, more prospective studies are needed to confirm our findings. Finally,

since the majority of Taiwan's population comes from Chinese backgrounds, the results of this study may not be generalizable to populations of different races.

## Conclusion

Our results suggest that TBI increases the risk of HF and CHD in this nationwide cohort of Taiwanese citizens. TBI has a modest but significant effect on the risk of HF and CHD. The incidence of CHD is higher than the incidence of HF. Men were likely to develop HF and women under 50 years of age are likely to develop CHD after TBI injury. Treatment of TBI with β-blockers offers a neutral effect on the risk of HF and CHD. However, patients with TBI who did not receive β-blockers treatment may increase the risk of HF.

## Supporting information

**S1 Checklist. STROBE statement—checklist of items that should be included in reports of *cohort studies.***
(DOC)

## Author Contributions

**Conceptualization:** Chao-Tung Yang, Chia-Chu Chang.

**Data curation:** Chia-Chu Chang.

**Formal analysis:** Ching-Hui Huang.

**Methodology:** Ching-Hui Huang, Chia-Chu Chang.

**Supervision:** Chao-Tung Yang, Chia-Chu Chang.

**Validation:** Chao-Tung Yang, Chia-Chu Chang.

**Writing – original draft:** Ching-Hui Huang.

**Writing – review & editing:** Chao-Tung Yang.

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
