## [Decision Letter · Decision Letter 0]

23 Aug 2023

PONE-D-23-22482Traumatic brain injury and risk of heart failure and coronary heart disease: A nationwide population-based cohort studyPLOS ONE

Dear Dr. Chang,

Thank you for submitting your manuscript to PLOS ONE. After careful consideration, we feel that it has merit but does not fully meet PLOS ONE’s publication criteria as it currently stands. Therefore, we invite you to submit a revised version of the manuscript that addresses the points raised during the review process.

The manuscript needs to be revised according to the Reviewers' suggestions. Respond to them appropriately.

We look forward to receiving your revised manuscript.

Kind regards,

Masaki Mogi

Academic Editor

PLOS ONE

2. Please include a separate caption for each figure in your manuscript.

Reviewers' comments:

Reviewer's Responses to Questions

**Comments to the Author**

1. Is the manuscript technically sound, and do the data support the conclusions?

Reviewer #1: Yes

Reviewer #2: Partly

2. Has the statistical analysis been performed appropriately and rigorously? 

Reviewer #1: Yes

Reviewer #2: Yes

3. Have the authors made all data underlying the findings in their manuscript fully available?

Reviewer #1: Yes

Reviewer #2: Yes

4. Is the manuscript presented in an intelligible fashion and written in standard English?

Reviewer #1: Yes

Reviewer #2: Yes

5. Review Comments to the Author

Reviewer #1: I appreciate the authors presenting this research article. My comments are as follows

1. The major drawback of current study was the discussion was not thorough enough. The authors repeated the results several times during the discussion. The discussion should focus on the reasons and possible explanations for the differences or similarities between the results and the previous studies. For example, why is there a high risk of heart failure without the use of beta-blockers?

2. The authors suggest The mechanisms for CHD in TBI patients were complex, possible related to disruption of autonomic regulation post TBI, or neuroinflammation induced systemic inflammation, or psychiatric

comorbidities post TBI, which may increase the risk of CHD. This is too general a statement and should be explained in more depth

3. The source of the data is recorded from 1996 to 2012, which is 10 years ago. The authors should discuss how the results contribute to current healthcare.

Reviewer #2: This interesting paper follows a large cohort of adults in a Taiwanese health insurance data base to examine the risk of heart failure (HF) and coronary heart disease (CHD) after TBI and the impact of gender on that association. The authors are commended for this insightful use of this large sample to address an important and clinically relevant question. However, the results are over-interpreted and the limitation of missing TBI severity information needs to be addressed throughout the paper and not just in the limitations section. Therefore, a major rewrite would be required before publication could be conisdered.

Strengths:

Results are very well-written.

Statistical methodology, particularly 1:4 matching using propensity scores, is strong. Methodology and decision-making are very well explicated.

Questions/concerns:

Intro refers to severe TBI and neurogenic cardiomyopathy- were only pts with severe brain injuries included?

Given the (significant) limitation that TBI severity was unavailable, the intro discussing severe TBI and neurogenic cardiomyopathy seems to have limited relevance. Would recommend we-writing intro and discussion and putting the study and its results in the context of the larger literature on TBI and cardiovascular disease.

concern about excluding prior TBIs- how many people were excluded for this reason? Study flowchart needs more detail.

why was age 40 selected as cutpoint for the subgroup analyses?

Adjusted effects of TBI on HF and CHD are quite small. A p value of 0.03 in a sample with an N of hundreds of thousands is not very impressive- was using a more strict p value cutoff considered? Given the sample size the results seem over-interpreted to me.

I would recommend removing the results/numerical values from the discussion section and keeping those in the results.

Discussion could benefit from English language editing.

More info on data availability is needed- who can access the data set? How?

6. PLOS authors have the option to publish the peer review history of their article (what does this mean?). If published, this will include your full peer review and any attached files.

Reviewer #1: No

Reviewer #2: No

---

## [Author Response · Author response to Decision Letter 0]

17 Oct 2023

Dear Academic Editor/ Professor Masaki Mogi, 

Thank you very much for your reviewing process of our manuscript “Traumatic brain injury and risk of heart failure and coronary heart disease: A nationwide population-based cohort study” (PONE-D-23-22482). The comments raised by the reviewers and Academic Editor were helpful and have been integrated into this revised submission. We appreciate the editor’s and reviewers’ comments to improve the readability of the manuscript; each of their points has been addressed. Revised portions are highlighted in red in the revised manuscript.

The followings are our point-to-point responses to the comments:

Elucidations for Reviewer 1:

Comment1-1 (C 1-1): The major drawback of current study was the discussion was not thorough enough. The authors repeated the results several times during the discussion. The discussion should focus on the reasons and possible explanations for the differences or similarities between the results and the previous studies. For example, why is there a high risk of heart failure without the use of beta-blockers? 

Response 1-1(R 1-1): Thank you for your valuable comment and reminders. We rewrote the discussion. The revised discussion is as follows:

 The incidence of CHF was higher in the TBI cohort than in the control cohort During the follow-up period, the number of HF events and the cumulative incidence of HF were significantly higher than in the control cohort (Table 2, Figure 2). The association of TBI with subsequent CHF was not attenuated in multivariate models, suggesting that TBI may account for risk independent of the other variables. Our findings highlight some important points. First, to our best knowledge, our study provided general population data on the incidence of HF following TBI insults. The incidence was lower than that of CHD. Previous large-scale cohort studies focused on composite end points for cardiovascular disease (coronary artery disease, stroke, and peripheral artery disease) in U.S Veterans after 9/11 [5], or with the subsequent incidence of major adverse cardiovascular and cerebrovascular events (MACCE) in the general population. This definition of MACCE include a composite of cardiovascular disease, ischemic stroke, and death [15]. No large-scale study had mentioned the incidence of HF following TBI events. Most patients with stress cardiomyopathy show regression of cardiac function independently of cardiac intervention, ranging from hours to 6-12 weeks [16]. However, patients with stress cardiomyopathy remain at risk of recurrence even years after the initial event [17]. Our data provided information to the knowledge gap to better understand the interaction between TBI and HF development. Second, a long-standing belief is that older women have a higher prevalence of stress cardiomyopathy [4,16]. However, in our study we found that the risk of incident HF was significantly higher in men than women in the population of TBI, and the mean age of the TBI cohort was 42.56±18.2 years old. Our real-world data contrary to previous belief. Third, previous meta-analyses have shown that beta-blockers are effective in reducing mortality in patients with TBI [10]. Exposure to beta-blockers appears to improve functional outcome in patients with severe TBI in a matched case-control study [18]. However, it is unclear whether the beta-blockers used in patients with TBI reduce the risk of subsequent HF. In our study, we found that treatment of patients with β-blockers offers a neutral effect on the risk of HF and CHD, but patients with TBI who did not receive β-blockers treatment may increase the risk of HF. Our result suggested that the beta-blocker used after TBI might have a beneficial effect on reducing the risk of subsequent incidence of HF. Because this database is a claim database, we were unable to explore the reasons why patients should or should not use beta-blockers after TBI. 

Age Stratification and Risk of CHD after TBI To determine whether the risk of CHD and HF changes with age, we assessed the risk after TBI compared to those younger than or older than 40 years. The 40-year cutoff point was based on previous large cohort studies [6,19], both of which showed that younger age groups (18–40 years) had significantly higher risk of cardiovascular disease. Our results indicate an association between TBI and CHD in all age groups without a baseline diagnosis. The risk of CHD was significantly higher in the age group less than 40 years, with an adjusted hazard ratio of 1.30. Our findings are consistent with previous studies showing that TBI of varying severity increases the risk of subsequent CHD, and the risk is significantly higher in patients younger than 40 years of age [6,19]. Regarding gender, assuming menopause at 50 years old, we found that women with TBI under 50 years old had a significantly higher risk of CHD, which is contrary to what we have long known about the incidence of CHD in menopausal women. Our data are consistent with previous studies showing that TBI is a potential risk factor for CHD even in premenopausal women [5,6,19,20].

Mechanisms of CHD in patients with TBI were complex. An observational study has shown the presence of coronary artery calcification in patients with mild TBI [20]. A preclinical study in mice supports a causal relationship between TBI and atherosclerosis [21], which may be related to increased sympathetic activity. Studies in rodents and humans have found long-lasting increases in sympathetic activity after TBI [22-24]. Beta-adrenergic receptor antagonism reduces accelerated atherosclerosis after TBI, as demonstrated in a mouse model deficient in apolipoprotein E [25]. Sympathoadrenal activation after TBI induces coagulopathy and endotheliopathy, as demonstrated by biomarkers, and these effects are associated with a poor prognosis [26]. A previous study has demonstrated the effects of TBI on leukocyte activation and endothelial adhesion biomarkers [21], although the exact mechanisms of accelerated atherosclerosis remain to be demonstrated. In addition to the systemic effects of catecholamines on circulating monocytes or endothelial cells after brain injury, efferent sympathetic peripheral nervous system axons may be stimulated to produce epinephrine locally in susceptible arteries, leading to increased inflammatory plaque activity [27]. All of the above possible explanations may explain why patients with TBI may be at increased risk for developing CHD. TBI severity and subsequent CHD and HF risks Current tools to assess the severity of TBI include the level of consciousness, typically assessed using the Glasgow Coma Scale (GCS), and the duration of posttraumatic amnesia [28]. The use of GCS scores is more commonly used. TBI can be classified into mild, moderate, and severe injury types based on patient scores reported using the GCS [29]. Mild traumatic brain injuries can result from concussions during sports activities, falls in elderly subjects, and battlefield injuries; while moderate to severe TBI are responsible for the majority of death and disability [29]. The incidence of mild TBI has been reported to be higher than that of moderate to severe TBI [30]. In terms of cardiovascular comorbidities, mild TBI (e.g. concussion) increases the risk of CHD even in younger age groups [19]. Furthermore, in a longitudinal study of individuals with TBI up to 10 years after injury, the authors found that CHD was highly prevalent in subjects with mild TBI, as well as moderate to severe injuries, and that some cardiovascular disease risk was greater in subjects with mild TBI [6]. Data are inconsistent when it comes to the risk of HF and the severity of TBI. A study showed that the severity of head computed tomography images was not associated with the development of cardiac dysfunction [31]. Another observational study showed that traumatic brain injury was not associated with significant myocardial dysfunction [32]. However, a study has reported that previously healthy patients with moderate to severe traumatic brain injury have a higher incidence of early systolic dysfunction than patients with mild traumatic brain injury and that it is reversible within the first week of hospitalization [33]. In our study, we did not emphasize the severity of TBI and the risk of CHD or HF in the absence of consistent arguments. We aimed to use this national database to simultaneously assess long-term, CHD, and HF events in a large and representative sample of patients with TBI compared to a matched cohort. 

Clinical implication A major focus of TBI management is to limit secondary brain damage. However, some patients with TBI develop extracranial multi-organ dysfunction, leading to secondary brain injury, increased risk of death, and poor functional prognosis after TBI. Although the data source records from 1996 to 2012, that is, 10 years ago. Our study can provide information on the incidence of these interactions (i.e. HF and CHD after TBI) to fill the knowledge gap resulting from the lack of randomized clinical trials and identify susceptible subgroups for screening and intervention to improve the prognosis of the patient. Increase the awareness of the healthcare provider and patient of TBI as a potential risk factor for CHD and HF, highlighting that aggressive control of cardiovascular risk factors is mandatory, especially in young adults after TBI. 

C 1-2: The authors suggest The mechanisms for CHD in TBI patients were complex, possible related to disruption of autonomic regulation post TBI, or neuroinflammation induced systemic inflammation, or psychiatric comorbidities post TBI, which may increase the risk of CHD. This is too general a statement and should be explained in more depth 

R 1-2: Thank you for your important comments. We have added paragraphs to the discussion section to provide insight into possible explanations. The revised manuscript as follows: 

In Discussion section Age Stratification and Risk of CHD after TBI 

….. “Mechanisms of CHD in patients with TBI were complex. An observational study has shown the presence of coronary artery calcification in patients with mild TBI [20]. A preclinical study in mice supports a causal relationship between TBI and atherosclerosis [21], which may be related to increased sympathetic activity. Studies in rodents and humans have found long-lasting increases in sympathetic activity after TBI [22-24]. Beta-adrenergic receptor antagonism reduces accelerated atherosclerosis after TBI, as demonstrated in a mouse model deficient in apolipoprotein E [25]. Sympathoadrenal activation after TBI induces coagulopathy and endotheliopathy, as demonstrated by biomarkers, and these effects are associated with a poor prognosis [26]. A previous study has demonstrated the effects of TBI on leukocyte activation and endothelial adhesion biomarkers [21], although the exact mechanisms of accelerated atherosclerosis remain to be demonstrated. In addition to the systemic effects of catecholamines on circulating monocytes or endothelial cells after brain injury, efferent sympathetic peripheral nervous system axons may be stimulated to produce epinephrine locally in susceptible arteries, leading to increased inflammatory plaque activity [27]. All of the above possible explanations may explain why patients with TBI may be at increased risk for developing CHD. ”

C1-3: The source of the data is recorded from 1996 to 2012, which is 10 years ago. The authors should discuss how the results contribute to current healthcare.

R 1-3: Thanks for the important comment. We have addressed this question in the discussion paragraph. In Discussion section Clinical implication A major focus of TBI management is to limit secondary brain damage. However, some patients with TBI develop extracranial multi-organ dysfunction, leading to secondary brain injury, increased risk of death, and poor functional prognosis after TBI. Although the data source records from 1996 to 2012, that is, 10 years ago. Our study can provide information on the incidence of these interactions (i.e. HF and CHD after TBI) to fill the knowledge gap resulting from the lack of randomized clinical trials and identify susceptible subgroups for screening and intervention to improve the prognosis of the patient. Increase the awareness of the healthcare provider and patient of TBI as a potential risk factor for CHD and HF, highlighting that aggressive control of cardiovascular risk factors is mandatory, especially in young adults after TBI. 

Elucidations for Reviewer 2:

Comment 2-1 (C 2-1): This interesting paper follows a large cohort of adults in a Taiwanese health insurance data base to examine the risk of heart failure (HF) and coronary heart disease (CHD) after TBI and the impact of gender on that association. The authors are commended for this insightful use of this large sample to address an important and clinically relevant question. However, the results are over-interpreted and the limitation of missing TBI severity information needs to be addressed throughout the paper and not just in the limitations section. Therefore, a major rewrite would be required before publication could be conisdered.

Response 2-1 (R 2-1): Thank you for your valuable comment and reminders. We rewrote the introduction and discussion section. We have added paragraphs to the Discussion and Limitations sections to discuss the severity of TBI and subsequent risk of CHD and HF. The revised discussion is as follows: 

Discussion: 

TBI severity and subsequent CHD and HF risks

Current tools to assess the severity of TBI include the level of consciousness, typically assessed using the Glasgow Coma Scale (GCS), and the duration of posttraumatic amnesia [28]. The use of GCS scores is more commonly used. TBI can be classified into mild, moderate, and severe injury types based on patient scores reported using the GCS [29]. Mild traumatic brain injuries can result from concussions during sports activities, falls in elderly subjects, and battlefield injuries; while moderate to severe TBI are responsible for the majority of death and disability [29]. The incidence of mild TBI has been reported to be higher than that of moderate to severe TBI [30]. In terms of cardiovascular comorbidities, mild TBI (e.g. concussion) increases the risk of CHD even in younger age groups [19]. Furthermore, in a longitudinal study of individuals with TBI up to 10 years after injury, the authors found that CHD was highly prevalent in subjects with mild TBI, as well as moderate to severe injuries, and that some cardiovascular disease risk was greater in subjects with mild TBI [6]. Data are inconsistent when it comes to the risk of HF and the severity of TBI. A study showed that the severity of head computed tomography images was not associated with the development of cardiac dysfunction [31]. Another observational study showed that traumatic brain injury was not associated with significant myocardial dysfunction [32]. However, a study has reported that previously healthy patients with moderate to severe traumatic brain injury have a higher incidence of early systolic dysfunction than patients with mild traumatic brain injury and that it is reversible within the first week of hospitalization [33]. In our study, we did not emphasize the severity of TBI and the risk of CHD or HF in the absence of consistent arguments. We aimed to use this national database to simultaneously assess long-term, CHD, and HF events in a large and representative sample of patients with TBI compared to a matched cohort.

In study limitations

“…Second, NHIRD did not provide some information, such as smoking history, body mass index, physical activity, blood pressure, inflammatory markers, blood sugar, blood lipids, and TBI severity. Lacking information on TBI severity, we were unable to measure whether there were dose-dependent effects on severity and subsequent risk of CHD or HF. These unmeasured covariates may have influenced the results, even after balancing baseline clinical characteristics and propensity score matching.”

C 2-2: Strengths:

Results are very well-written. Statistical methodology, particularly 1:4 matching using propensity scores, is strong. Methodology and decision-making are very well explicated.

R 2-2: Thank you for your comment. 

C 2-3: Intro refers to severe TBI and neurogenic cardiomyopathy- were only pts with severe brain injuries included? Given the (significant) limitation that TBI severity was unavailable, the intro discussing severe TBI and neurogenic cardiomyopathy seems to have limited relevance. Would recommend re-writing intro and discussion and putting the study and its results in the context of the larger literature on TBI and cardiovascular disease.

R 2-3: Thank you for your important comment. We rewrote the introduction and discussion, and focused on brain-heart interactions. We explore the issue of TBI severity in the Discussion section, subtitled “TBI severity and subsequent CHD and HF risk,” and also address this issue in the Limitations section. The revised manuscript as follows:

Introduction: 

“Traumatic brain injury (TBI) is a serious public health problem worldwide associated with an increased risk of disability and mortality [1]. Brain-heart interactions are most common in traumatic brain injury and manifest as arrhythmias, neurogenic myocardial stunning or stress cardiomyopathy, hemodynamic disturbances, and death [2,3]. In most patients with stress cardiomyopathy, cardiac function may return within hours to 6-12 weeks, with or without cardiac intervention. However, patients with stress cardiomyopathy remain at risk of recurrence even years after the initial event [4] and there are limited data on the incidence of these interactions (i.e., heart failure after TBI). Previous studies have focused on associations with the risk of coronary heart disease (CHD) after TBI, while little information is available on the incidence of heart failure (HF) after TBI [5,6]. Few studies have simultaneously investigated CHD and HF after TBI. Furthermore, women are prone to stress cardiomyopathy [7]. However, the long-term follow-up risks of HF and CHD by gender in patients with TBI are unknown.

 Brain-heart crosstalk, which occurs due to stress-induced surges in catecholamines after brain injury and brain injury-induced release of inflammatory mediators, can further damage cardiomyocytes and affect patient clinical outcomes [8,9]. Since sympathetic system overactivity is believed to play a key role in the cardiac manifestations of TBI, several studies have investigated the protective role of beta-blockers in this setting [3]. A recent meta-analysis demonstrated that beta-blocker use after TBI reduces in-hospital mortality [10]. However, it is unclear whether the use of beta-blockers in patients with TBI reduces the risk of subsequent HF. 

 A better understanding of the interaction between TBI and the development of cardiovascular comorbidities may have important implications for preventive care, prognosis, and targeted screening of high-risk populations. The purpose of our study was to evaluate whether the incidence of CHD and HF is higher in a cohort of TBI than in controls and whether there are gender differences. Furthermore, it was explored whether the use of beta-blockers in patients with TBI may reduce the risk of subsequent HF.” 

Discussion: 

“TBI severity and subsequent CHD and HF risks

Current tools to assess the severity of TBI include the level of consciousness, typically assessed using the Glasgow Coma Scale (GCS), and the duration of posttraumatic amnesia [28]. The use of GCS scores is more commonly used. TBI can be classified into mild, moderate, and severe injury types based on patient scores reported using the GCS [29]. Mild traumatic brain injuries can result from concussions during sports activities, falls in elderly subjects, and battlefield injuries; while moderate to severe TBI are responsible for the majority of death and disability [29]. The incidence of mild TBI has been reported to be higher than that of moderate to severe TBI [30]. In terms of cardiovascular comorbidities, mild TBI (e.g. concussion) increases the risk of CHD even in younger age groups [19]. Furthermore, in a longitudinal study of individuals with TBI up to 10 years after injury, the authors found that CHD was highly prevalent in subjects with mild TBI, as well as moderate to severe injuries, and that some cardiovascular disease risk was greater in subjects with mild TBI [6]. Data are inconsistent when it comes to the risk of HF and the severity of TBI. A study showed that the severity of head computed tomography images was not associated with the development of cardiac dysfunction [31]. Another observational study showed that traumatic brain injury was not associated with significant myocardial dysfunction [32]. However, a study has reported that previously healthy patients with moderate to severe traumatic brain injury have a higher incidence of early systolic dysfunction than patients with mild traumatic brain injury and that it is reversible within the first week of hospitalization [33]. In our study, we did not emphasize the severity of TBI and the risk of CHD or HF in the absence of consistent arguments. We aimed to use this national database to simultaneously assess long-term, CHD, and HF events in a large and representative sample of patients with TBI compared to a matched cohort.” 

Study limitation 

“…Second, NHIRD did not provide some information, such as smoking history, body mass index, physical activity, blood pressure, inflammatory markers, blood sugar, blood lipids, and TBI severity. Lacking information on TBI severity, we were unable to measure whether there were dose-dependent effects on severity and subsequent risk of CHD or HF. These unmeasured covariates may have influenced the results, even after balancing baseline clinical characteristics and propensity score matching.…” 

C 2-4: concern about excluding prior TBIs- how many people were excluded for this reason? Study flowchart needs more detail.

R 2-4: Thank you for your important comment. TBI Cohort: Patients diagnosed with TBI between 1996 and 2012 (ICD-9-CM codes: 800, 801, 803, 804, 850-853, 854.1, and 959.01) were selected. “TBI” in this study specifically refers to patients newly diagnosed with traumatic brain injury, and the index date was determined as the initial date of TBI diagnosis. Therefore, no patients with prior TBI were omitted from the study's participant pool.

The 2 million random beneficiary data in the original NHIRD database are currently provided by the Health and Welfare Data Center (HWDC), and the previous 1 million random beneficiary data from the original NHIRD database have expired. Unfortunately, due to the expiration of the database, we are unable to retrieve these 1 million random data sets again, nor can we provide a more detailed flowchart. But the main information is already contained in the previous flow chart.

C 2-5: why was age 40 selected as cutpoint for the subgroup analyses?

R 2-5: Thank you for your important comment. We selected age 40 as the cut-off point based on previous studies [Ref. 6,19]. We discuss this issue in the Discussion section, subtitled “Age stratification and risk of CHD after TBI.” The revised discussion is as follows: 

“Age Stratification and Risk of CHD after TBI

To determine whether the risk of CHD and HF changes with age, we assessed the risk after TBI compared to those younger than or older than 40 years. The 40-year cutoff point was based on previous large cohort studies [6,19], both of which showed that younger age groups (18–40 years) had significantly higher risk of cardiovascular disease. Our results indicate an association between TBI and CHD in all age groups without a baseline diagnosis. The risk of CHD was significantly higher in the age group less than 40 years, with an adjusted hazard ratio of 1.30. Our findings are consistent with previous studies showing that TBI of varying severity increases the risk of subsequent CHD, and the risk is significantly higher in patients younger than 40 years of age [6,19]. …”

 Ref.6- Izzy S, Chen PM, Tahir Z, Grashow R, Radmanesh F, Cote DJ, et al. Association of traumatic brain injury with the risk of developing chronic cardiovascular, endocrine, neurological, and psychiatric disorders. JAMA network open. 2022;5(4):e229478-e.

 Ref.19- Izzy S, Tahir Z, Grashow R, Cote DJ, Jarrah AA, Dhand A, et al. Concussion and risk of chronic medical and behavioral health comorbidities. Journal of neurotrauma. 2021;38(13):1834-41.

C 2-6: Adjusted effects of TBI on HF and CHD are quite small. A p value of 0.03 in a sample with an N of hundreds of thousands is not very impressive- was using a more strict p value cutoff considered? Given the sample size the results seem over-interpreted to me.

R 2-6: Thank you for your important comment. You are correct in noting that with a very large sample size (in the hundreds of thousands), even very small deviations from the null hypothesis can lead to statistically significant p-values. In such cases, it's important to consider not only the p-value but also the practical significance or effect size of the findings. For example, for heart failure in our study, the hazard ratios and adjusted hazard ratios for heart failure risk in the TBI cohort were 1.13, 1.11, and 1.08, respectively, in models 1–3. Although the effect was modest, clinically it provides clues that TBI is a potential risk factor for heart failure. This is the advantage of large samples in detecting and quantifying small or complex effects. In studies with small sample sizes, we were unable to detect small but important effects. As you mentioned, some researchers have different recommendations for ever-decreasing p-values: adjust the threshold p-value downward as sample size increases. The argument is that, for example, for very large samples, the threshold should be 1%, 0.1% or even smaller, rather than claiming significance at p < 5%. However, this approach has not yet proposed empirical rules for how to make such adjustments (Reference #). There is no universal answer to this question. In our study, we not only focus on P values but also report confidence intervals and effect sizes (hazard ratios) to give readers a complete understanding of the data.

 Reference#- Lucas, H. and G. Shmueli, Too big to fail: large samples and the p-value problem. Inf. Syst. Res., 2013. 24(4): p. 906-917.

C 2-7: I would recommend removing the results/numerical values from the discussion section and keeping those in the results.

R 2-7: Thank you for your important comment. We have removed the results/numerical values from the discussion section and keeping those in the results.

C 2-8: Discussion could benefit from English language editing.

R 2-8: Thank you for your reminder. Discussion section has been modified by a native English speaker.

C 2-9: More info on data availability is needed- who can access the data set? How?

R 2-9: Thank you for raising this important question. Each applicant seeking to use the National Health Insurance Research Database (NHIRD) must be a researcher or clinician from a university, institute, or hospital, and use of the data must be for research purposes only. All applications are subject to expert review to ensure reasonable use. Applicants must obtain IRB approval before applying to access the NHIRD. Currently, only Taiwanese researchers have direct access to the data. Recently, Taiwan’s Ministry of Health and Welfare (MOHW) established a Health and Welfare Data Center (HWDC), a data repository site that centralizes the NHIRD databases for data management and analyses. To strengthen the protection of data privacy, investigators are required to conduct on-site analysis at an HWDC through remote connection to MOHW servers.

We hope you will find our revised manuscript to be satisfactory and suitable for publication in your journal, PLOS ONE. 

Thank you very much,

With best regard

Chia-Chu Chang, MD, PhD

Department of Internal Medicine, Kuang Tien General Hospital, Taichung, Taiwan. 

No.117, Shatian Road Shalu District, Taichung City 433, Taiwan 

E-mail: Chia-Chu Chang <chiachuchang0312@gmail.com>

---

## [Decision Letter · Decision Letter 1]

2 Nov 2023

PONE-D-23-22482R1Traumatic brain injury and risk of heart failure and coronary heart disease: A nationwide population-based cohort studyPLOS ONE

Dear Dr. Chang,

Thank you for submitting your manuscript to PLOS ONE. After careful consideration, we feel that it has merit but does not fully meet PLOS ONE’s publication criteria as it currently stands. Therefore, we invite you to submit a revised version of the manuscript that addresses the points raised during the review process.

The manuscript is necessary to be minor-revisioned according to the Reviewer's comments.

We look forward to receiving your revised manuscript.

Kind regards,

Masaki Mogi

Academic Editor

PLOS ONE

Journal Requirements:

Reviewers' comments:

Reviewer's Responses to Questions

**Comments to the Author**

1. If the authors have adequately addressed your comments raised in a previous round of review and you feel that this manuscript is now acceptable for publication, you may indicate that here to bypass the “Comments to the Author” section, enter your conflict of interest statement in the “Confidential to Editor” section, and submit your "Accept" recommendation.

Reviewer #1: All comments have been addressed

Reviewer #2: (No Response)

2. Is the manuscript technically sound, and do the data support the conclusions?

Reviewer #1: Yes

Reviewer #2: Yes

3. Has the statistical analysis been performed appropriately and rigorously? 

Reviewer #1: (No Response)

Reviewer #2: Yes

4. Have the authors made all data underlying the findings in their manuscript fully available?

Reviewer #1: No

Reviewer #2: No

5. Is the manuscript presented in an intelligible fashion and written in standard English?

Reviewer #1: Yes

Reviewer #2: (No Response)

6. Review Comments to the Author

Reviewer #1: The revised manuscript hae replied my comments. I have no more comments. Accept is my final decision.

Reviewer #2: This reviewer wishes to thank the authors for their thoughtful and comprehensive response to the reviewer comments. This paper is now acceptable for publication, pending a few minor changes: there are several places (ie the end of the abstract) where the results are overstated/presented as factual statements: "TBI increases the risk of HF and CHD." All of these statements should be edited to say something like "Our results suggest that TBI increases the risk... in this nationwide cohort of Taiwanese citizens" or similar.

My only other concern is that the data do not appear to be publicly available (ie available only to Taiwanese citizens), which seems to be inconsistent with journal policy- that is for the editor to address.

7. PLOS authors have the option to publish the peer review history of their article (what does this mean?). If published, this will include your full peer review and any attached files.

Reviewer #1: No

Reviewer #2: No

---

## [Author Response · Author response to Decision Letter 1]

15 Nov 2023

Dear Academic Editor/ Professor Masaki Mogi, 

Thank you very much for your reviewing process of our manuscript “Traumatic brain injury and risk of heart failure and coronary heart disease: A nationwide population-based cohort study” (PONE-D-23-22482R1). The comments raised by the reviewers and Academic Editor were helpful and have been integrated into this revised submission. We appreciate the editor’s and reviewers’ comments to improve the readability of the manuscript; each of their points has been addressed. Revised portions are highlighted in red in the revised manuscript.

The followings are our point-to-point responses to the comments:

Elucidations for Reviewer 1:

Comment 1: The revised manuscript has replied my comments. I have no more comments. Accept is my final decision. 

Response 1: We appreciate the reviewers’ comments and thank you again.

Elucidations for Reviewer 2:

Comment 2: This reviewer wishes to thank the authors for their thoughtful and comprehensive response to the reviewer comments. This paper is now acceptable for publication, pending a few minor changes: there are several places (ie the end of the abstract) where the results are overstated/presented as factual statements: "TBI increases the risk of HF and CHD." All of these statements should be edited to say something like "Our results suggest that TBI increases the risk... in this nationwide cohort of Taiwanese citizens" or similar.

Response 2-1 (R 2-1): Thank you for your valuable comment and reminders. We followed your suggestion and rewrote it in a more conservative way in Abstract and Manuscript conclusion. The revised Abstract is as follows: 

Abstract: 

Background: This study examined the long-term risks of heart failure (HF) and coronary heart disease (CHD) following traumatic brain injury (TBI), focusing on gender differences.

Methods: Data from Taiwan's National Health Insurance Research Database included 29,570 TBI patients and 118,280 matched controls based on propensity scores.

Results: The TBI cohort had higher incidences of CHD and HF (9.76 vs. 9.07 per 1000 person-years; 4.40 vs. 3.88 per 1000 person-years). Adjusted analyses showed a significantly higher risk of HF in the TBI group (adjusted hazard ratio = 1.08, 95% CI = 1.01-1.17, P = 0.031). The increased CHD risk in the TBI cohort became insignificant after adjustment. Subgroup analysis by gender revealed higher HF risk in men (aHR = 1.14, 95% CI = 1.03-1.25, P = 0.010) and higher CHD risk in women under 50 (aHR = 1.32, 95% CI = 1.15-1.52, P < 0.001). TBI patients without beta-blocker therapy may be at increased risk of HF.

Conclusion: Our results suggest that TBI increases the risk of HF and CHD in this nationwide cohort of Taiwanese citizens. Gender influences the risks differently, with men at higher HF risk and younger women at higher CHD risk. Beta-blockers have a neutral effect on HF and CHD risk. TBI severity and subsequent CHD and HF risks

In manuscript conclusion

Our results suggest that TBI increases the risk of HF and CHD in this nationwide cohort of Taiwanese citizens. TBI has a modest but significant effect on the risk of HF and CHD. The incidence of CHD is higher than the incidence of HF. Men were likely to develop HF and women under 50 years of age are likely to develop CHD after TBI injury. Treatment of TBI with β-blockers offers a neutral effect on the risk of HF and CHD. However, patients with TBI who did not receive β-blockers treatment may increase the risk of HF.

We hope you will find our revised manuscript to be satisfactory and suitable for publication in your journal, PLOS ONE. 

Thank you very much,

With best regard

Chia-Chu Chang, MD, PhD

Department of Internal Medicine, Kuang Tien General Hospital, Taichung, Taiwan. 

No.117, Shatian Road Shalu District, Taichung City 433, Taiwan 

E-mail: Chia-Chu Chang <chiachuchang0312@gmail.com>

---

## [Decision Letter · Decision Letter 2]

21 Nov 2023

Traumatic brain injury and risk of heart failure and coronary heart disease: A nationwide population-based cohort study

PONE-D-23-22482R2

Dear Dr. Chang,

We’re pleased to inform you that your manuscript has been judged scientifically suitable for publication and will be formally accepted for publication once it meets all outstanding technical requirements.

Kind regards,

Masaki Mogi

Academic Editor

PLOS ONE

Additional Editor Comments (optional):

Reviewers' comments:

Reviewer's Responses to Questions

**Comments to the Author**

1. If the authors have adequately addressed your comments raised in a previous round of review and you feel that this manuscript is now acceptable for publication, you may indicate that here to bypass the “Comments to the Author” section, enter your conflict of interest statement in the “Confidential to Editor” section, and submit your "Accept" recommendation.

Reviewer #2: All comments have been addressed

2. Is the manuscript technically sound, and do the data support the conclusions?

Reviewer #2: Yes

3. Has the statistical analysis been performed appropriately and rigorously? 

Reviewer #2: Yes

4. Have the authors made all data underlying the findings in their manuscript fully available?

Reviewer #2: No

5. Is the manuscript presented in an intelligible fashion and written in standard English?

Reviewer #2: Yes

6. Review Comments to the Author

Reviewer #2: Thank you for your careful revision. Happy to recommend publication of this paper in its current form.

7. PLOS authors have the option to publish the peer review history of their article (what does this mean?). If published, this will include your full peer review and any attached files.

Reviewer #2: No

---

## [Editor Report · Acceptance letter]

28 Nov 2023

PONE-D-23-22482R2 

Traumatic brain injury and risk of heart failure and coronary heart disease: A nationwide population-based cohort study 

Dear Dr. Chang:

I'm pleased to inform you that your manuscript has been deemed suitable for publication in PLOS ONE. Congratulations! Your manuscript is now with our production department. 

Kind regards, 

on behalf of

Dr. Masaki Mogi 

Academic Editor

PLOS ONE